# Phase biasing of a Josephson junction using Rashba–Edelstein effect

Tapas Senapati [1], Ashwin Kumar Karnad [2] & Kartik Senapati [1] ✉

A charge-current-induced shift in the spin-locked Fermi surface leads to a non-equilibrium spin density at a Rashba interface, commonly known as the Rashba–Edelstein effect. Since this is an intrinsically interfacial property, direct detection of the spin moment is difficult. Here we demonstrate that a planar Josephson Junction, realized by placing two closely spaced superconducting electrodes over a Rashba interface, allows for a direct detection of the spin moment as an additional phase in the junction. Asymmetric Fraunhofer patterns obtained for Nb-(Pt/Cu)-Nb nano-junctions, due to the locking of Rashba–Edelstein spin moment to the flux quantum in the junction, provide clear signatures of this effect. This simple experiment offers a fresh perspective on direct detection of spin polarization induced by various spin-orbit effects. In addition, this platform also offers a magnetic-field-controlled phase biasing mechanism in conjunction with the Rashba–Edelstein spin-orbit effect for superconducting quantum circuits.

Spin-orbit coupling has emerged as a clean process of generating pure spin current and spin polarization in solid-state spintronics, without using a magnetic layer[1-4]. The effective magnetic field arising from the drift motion of an electron in the radial atomic potential gradient of a bulk metallic system decouples the motion of spin-up and spin-down electrons, leading to a non-equilibrium spin current in a direction transverse to the charge current, popularly known as spin-Hall effect[5-7]. The Onsager reciprocity relation also allows for an inverse effect, where a bulk spin current leads to a charge drift current in a transverse direction[8,9]. A related effect ensues dominantly at an interface with broken structural inversion symmetry due to the interfacial potential gradient. In this case, electron spin locks to the crystal momentum through the Rashba Hamiltonian ($H_R = \frac{\alpha_R}{\hbar}(E_z \times p) \cdot S$) and the up/down spin bands split in energy[10-12]. Therefore, a charge current driven by an external electric field translates the parabolic band leading to a net spin polarization[10-12]. In the real space scenario, a charge current parallel to a Rashba-interface (orthogonal to the interfacial electric field) leads to a non-equilibrium spin density orthogonal to the interfacial electric field and to the drift current following the relation ($S = \frac{\alpha_R m}{eh}(E_z \times j_c)$), where $E_z$ is the interfacial electric field and S is the spin polarization[13]. This inverse spin-galvanic effect or

Rashba–Edelstein effect has been studied in 2D electron gas and other metallic interfaces in proximity of heavy metals, especially in the context of spin-orbit torque devices[14-19]. At metal-metal interfaces, the interfacial electric field arises from the relative orbital alignment of the two metals in contact[20-23]. The basic protocol that all experimental probes followed is to measure a charge current produced by an injected spin polarization in a non-local measurement technique, which is the inverse Rashba–Edelstein effect[23-25]. The interaction of charge current with the spin density at the Rashba interface has also been studied via ferromagnetic resonance (FMR) techniques, by pumping microwaves through the interface[26-28].

Direct measurement of the charge current-induced surface spin density due to the Rashba–Edelstein effect is, however, challenging. In the case of spin-Hall effect, a spin current is induced by the charge current, which can carry a torque into a proximal ferromagnetic layer for detection. Rashba–Edelstein effect, on the other hand, only creates a non-equilibrium spin density (polarization) at the Rashba interface rather than a spin-current transferable to another layer for detection. The inverse Rashba–Edelstein effect and the FMR technique require a ferromagnetic layer for detection[29]. Here we report a phase-sensitive experiment enabling direct detection of this non-equilibrium spin

[1]School of Physical Sciences, National Institute of Science Education and Research (NISER) Bhubaneswar, An OCC of Homi Bhabha National Institute, Jatni 752050 Odisha, India. [2]Department of Physics, Birla Institute of Technology & Science Pilani - K K Birla Goa Campus, Zuarinagar 403726 Goa, India. ✉e-mail: kartik@niser.ac.in

density at the interface, without using a ferromagnetic layer. Figure 1 describes the basic architecture of the experiment. The Rashba interface was created by depositing a layer of Cu on top of a thin layer of Platinum[18,28,30]. As shown in the schematic Fig. 1a, a bias current across the Nb electrodes, in the normal state of Nb, is decomposed into parallel current channels through the Cu and the Pt layers, depicted as $J_{Cu}$ and $J_{Pt}$. Therefore, in the normal state of the Nb electrodes, $J_{Pt}$ produces a spin-Hall current ($J_S$) inside the Pt layer. Similarly, the current at the interface of the Cu and Pt layers (shown as $J_{Int}$ in Fig. 1a) produces a non-equilibrium spin density in the interfacial plane denoted as an equivalent moment $M_{RE}$. The bulk spin-Hall effect and the interface Rashba–Edelstein effects arise simultaneously in the system, which are difficult to isolate in real systems. On the other hand, sufficiently below the superconducting transition temperature of the Nb electrodes, a Josephson coupling can be established across the (Pt/Cu) barrier in the same device for a small enough separation between the Nb electrodes. In this planar Josephson Junction geometry, the bulk current in Pt can be entirely shorted through the proximatized Cu layer. This scenario is schematically shown in Fig. 1b. This simple geometry provides two immediate advantages, viz. (i) only a quasiparticle current can exist at the Cu/Pt Rashba-interface due to the pair breaking effects[31] which can generate the inverse spin-galvanic effect and (ii) the high phase sensitivity of the resulting planar Josephson Junction offers the possibility of direct sensing of the accumulated spin density[32], without having to transfer the spin-density information to another layer. Considering the geometry of the device in Fig. 1a, b, the Rashba field ($2\alpha_R k_F/g\mu_B$) points in the y-direction in the presence of a charge current in the x-direction. Here $\alpha_R$ is the Rashba parameter, $k_F$, g, and $\mu_B$ are the Fermi momentum, g-factor, and Bohr magnetron, respectively. The current driven relative displacement of the

momentum locked up-spin, and down-spin bands is shown in Fig. 1c, which leads to the Rashba–Edelstein effect. An external magnetic field applied perpendicular to any planar Josephson junction leads to the usual Fraunhofer-like critical current variations in the junction[33–35]. In the presence of an interface spin moment $M_{RE}$, in the Josephson device shown in Fig. 1b, a component of $M_{RE}$ can be coupled to the junction[36,37] which can show up in the Fraunhofer patterns. Here we show that indeed this is the case in Nb-(Pt/Cu)-Nb Josephson junctions(JJ) and SQUIDs.

## Results

### Josephson coupling across Nb-(Cu/Pt)-Nb planar junctions and nano-SQUIDs

The central results of this work have been shown in Fig. 2. Panel (a) shows the false colour electron micrograph of an actual DC nano-SQUID consisting of Nb-(Cu/Pt)-Nb planar junctions with 100 nm thick Cu and 50 nm thick Pt layer. In panel (b) of Fig. 2, we plot the voltage (at a constant current of 100 µAmps) across the nano-SQUID as an out-of-plane magnetic field was swept between ±20 mT. The resistance of the SQUID device shows characteristic oscillations superimposed on a Fraunhofer-like background, typically expected for a functional SQUID device. We note here that the field variation of the junction resistance is analogous to the field variation of critical current, and both quantities are related to each other in an inverse manner. We show the equivalence of both these measurements as a supplementary Fig. 2 and in Supplementary Fig. 6 by explicitly measuring the field variation of both quantities. Figure 2c shows the field variation of the device resistance for a single Josephson junction. In both cases (Fig. 2b, c), the Fraunhofer-like field response of the device resistance show a distinct offset between the forward and reverse sweeps of magnetic fields,

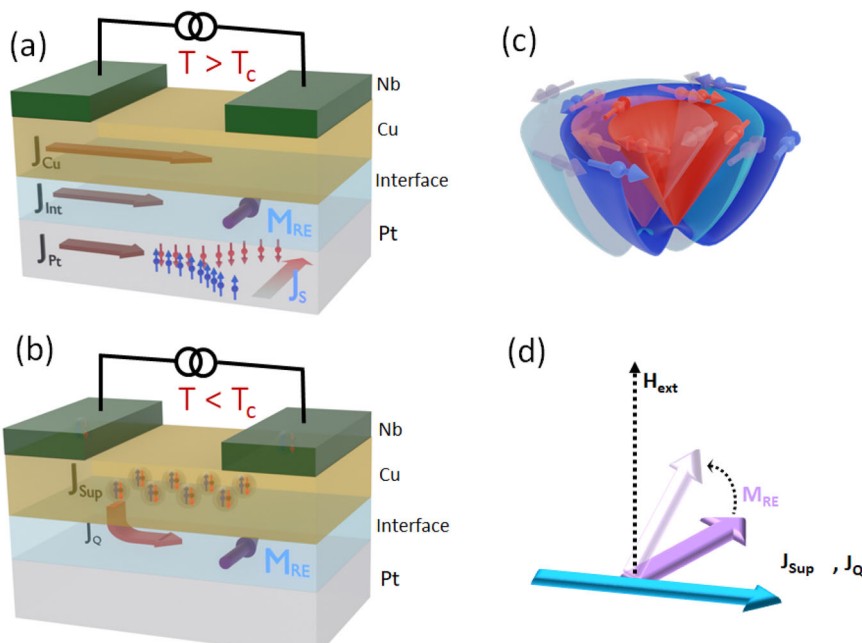

**Fig. 1 | Concept of interface spin polarization generation and detection in a planar Josephson junction. a** A schematic representation of the slitting of bias current in a Pt/Cu bilayer injected through Nb electrodes in the normal state of Nb. The total injected normal current, in this case, can be represented as a sum of the currents carried by the Pt layer ($J_{Pt}$), the Cu layers ($J_{Cu}$)and the Pt/Cu interface ($J_{Int}$). We have represented the Pt/Cu interface as a separate layer as the current at the interface produces a non-equilibrium spin moment $M_{RE}$ due to the Rashba–Edelstein effect. Current through the heavy metal Pt layer generates spin polarization in a transverse direction by the spin hall effect. **b** When the same device is cooled to a temperature below the transition temperature of Nb

electrodes ($T < T_c$), and a Josephson coupling is established between them, then the entire injected current is carried by the proximatized Cu layer. However, at the Pt/Cu interface, pair breaking by the spin-orbit coupling effects allows for some quasiparticle current $J_Q$. The Pt/Cu Rashba interface creates an in-plane spin polarization $M_{RE}$ due to the quasiparticle current $J_Q$. **c** The band representation of current driven shift of the momentum locked up-spin and down-spin bands causing a spin asymmetry at the Fermi surface. This causes the non-equilibrium spin moment depicted as $M_{RE}$ in panel (**a**, **b**). **d** In the presence of an external magnetic field the spin moment attains a component along the field which can couple into the junction area.

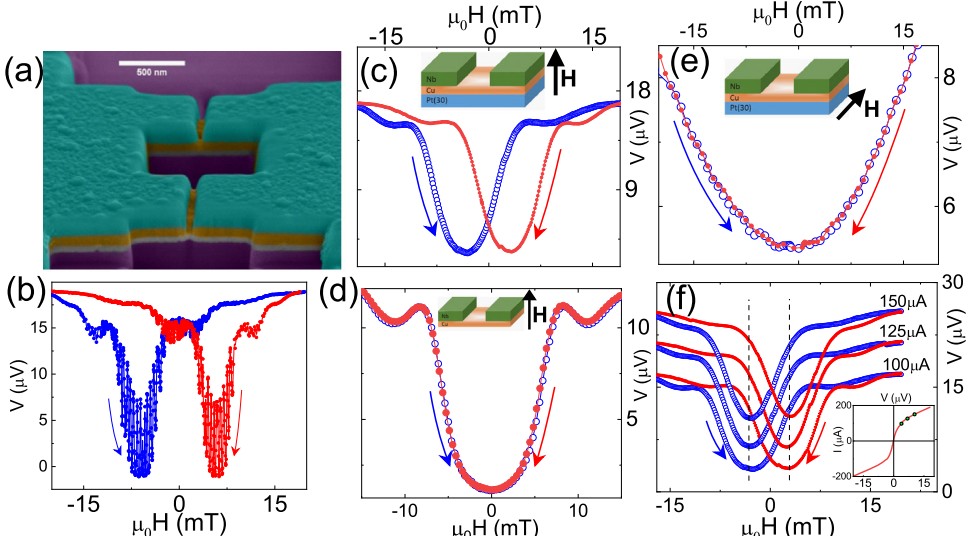

**Fig. 2 | Signature of interface Rashba–Edelstein effect in JJs and SQUIDs. a** False coloured SEM micrograph of an Nb-(Pt/Cu)-Nb SQUID, with 150 nm of Nb (green), 100 nm of Cu (orange), and 50 nm of Pt (uncoloured) (**b**) magnetic field response of the voltage across the same SQUID device, measured at 2K with a bias current equal to the critical current at that temperature. The external field was applied perpendicular to the plane of the loop. The arrows represent the direction of field sweep during these measurements. There is a clear offset between the up-sweep and down-sweep Fraunhofer patterns. **c** Magnetic field response of a single planar Josephson junction prepared from Nb-(Pt/Cu)-Nb trilayer with 30 nm of Pt layer also shows a similar offset between the up-sweep and down-sweep. On the contrary, the Fraunhofer pattern of a similar junction without the Pt layer does not show any relative shift between the up-sweep and down-sweep data in panel (**d**). **e** The Josephson device showing a clear offset in the Fraunhofer pattern in a perpendicular field (panel (**c**)) did not show any such shift in panel (**e**) when measured with an in-plane magnetic field. **f** The net offset in the Fraunhofer pattern for the Nb-(Pt/Cu)-Nb JJ with 30 nm Pt layer, measured at 2K with bias currents of 100 μA, 125 μA and 150 μA did not show any significant change. The inset in panel (**f**) shows the zero-field current-voltage(IV) curve and marks the bias currents on this plot.

which is unlike typical non-magnetic Josephson junctions[38–41]. In JJs with ferromagnetic barriers, hysteresis in the Fraunhofer pattern is observed arising from the flux remnance of the ferromagnetic barrier itself. During down sweep of the magnetic field from a positive saturation field, the remnant magnetic moment of a ferromagnetic barrier shifts the central maximum of the Fraunhofer pattern to the negative field, and vice-versa[42–45]. However, we would like to point out that the observed relative shift in the Fraunhofer pattern with voltage oscillation of the Nb-(Cu/Pt)-Nb SQUID in Fig. 2b is unlike the hysteresis in ferromagnetic JJs. In this case, while decreasing the magnetic field from the positive saturation, the central position of the Fraunhofer pattern appears to shift in the positive direction. The arrows in Fig. 2b, c indicate the direction of the field sweep in these measurements. In contrast, an Nb-Cu-Nb planar junction, without the Pt layer, prepared via the same route does not show any significant offset between the forward and reverse sweep of out-of-plane magnetic field, as shown in Fig. 2d. This observation clearly points towards the fact that the Cu/Pt interface is the primary reason behind the observed effect. Figure 2e plotted the voltage across the same junction when the magnetic field was applied parallel to the junction, as shown in the inset schematic. In contrast to the perpendicular field, the in-plane field does not induce any offset between the forward and reverse sweeps of the magnetic field. Another important characteristic of the observed offset in Fraunhofer pattern is that the junction bias current has a negligible effect on the net offset, as shown in Fig. 2f. This is consistent with the data obtained for SQUID devices also (see supplementary Fig. 5). The inset in Fig. 2f marks the bias currents used in these measurements with respect to the current-voltage characteristics of the same junction. In the next section, we argue that the non-equilibrium spin moment created by the Rashba–Edelstein effect at the interface between Pt and Cu introduces an additional phase in the Josephson junction, resulting in the unusual shift in the Fraunhofer pattern seen in Fig. 2.

## Generation of spin density at the Cu/Pt interface

The most prominent feature of Fig. 2 is the observation of a relative offset (ΔH) between the forward and reverse sweep of Fraunhofer patterns. Usually, the central peak of the Fraunhofer pattern of a standard JJ corresponds to a total of zero net flux in the junctions, equivalent to a constant phase difference of $\pi/2$ between the superconducting electrodes throughout the junction[46]. Therefore, a shift in the central peak can appear only in the presence of an additional phase in the junction. In ferromagnetic JJs, for example, the position of the central peak corresponds to the coercive fields where the net moment in the junction becomes zero[47,48]. In the present case, while sweeping the magnetic field from negative field, the central peak of the Fraunhofer pattern appears in the negative field regime. Since there are no magnetic moments in the present case, the observed shift can arise only from a net spin-moment present in the junction. Figure 2d shows that there is no relative shift in the Fraunhofer patterns in a junction without the Pt layer. Therefore, the presence noncentrosymmetric potential gradient in the Pt layer must be the origin of spin moment in the planar junction[49,50]. There are two ways in which a Pt layer can generate spin polarization. A normal current in the bulk of the Pt layer can generate a polarization via the spin-Hall effect[51–53]. Similarly, a current at the Pt/Cu interface can generate a spin-moment via Inverse spin-galvanic effect[30,54]. Therefore, in order to establish that the spin-moment seen by the JJs and SQUIDs in our case arises from the interface Rashba–Edelstein effect, the bulk spin-Hall effect must be ruled out.

For this purpose, we have performed the following control experiments. In our case, in the superconducting state of the planar junction, the Cu layer is proximatized by the superconducting electrodes and carries all the current[55]. This can be supported by the fact that the separation between the superconducting Nb electrodes in the planar JJs and SQUIDs used in this study varied between 40 nm and 100 nm for which the Pt layer underneath the Cu layer remains in the normal state down to the lowest measurement temperature.

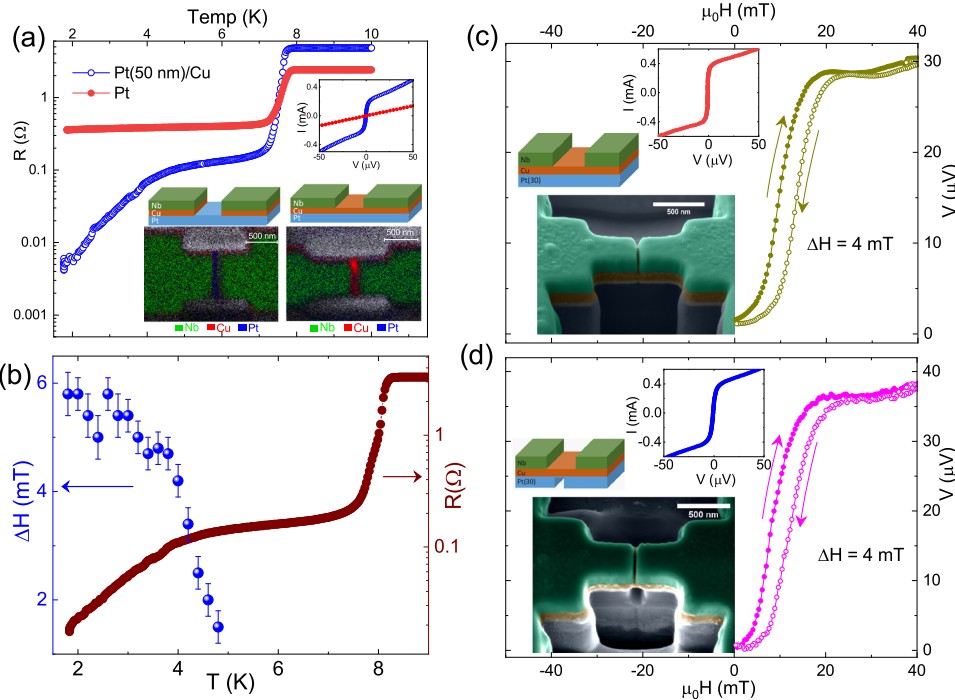

**Fig. 3 | Control experiments confirming a quasiparticle mediated Rashba−Edelstein effect. a** Comparison of low temperature resistance(R(T)) of an Nb(150 nm)-[Pt(50 nm)/Cu(100 nm)]-Nb(150 nm) JJ (open symbols) and Nb(150 nm)-Pt(50 nm)-Nb(150 nm) JJ(solid symbols)measured with a bias current of 10 µA. The junction without Cu-layer does not show any proximatisation. Inset shows the IV characteristics for both junctions at zero field. Nb-(Pt/Cu)-Nb junction shows a critical current ~200 µA. The inset EDAX elemental maps for both these junctions clearly show the absence of Cu in the Nb-Pt-Nb junction, as represented in the device schematics. **b** R(T) plot for an Nb-(Pt(30 nm)/Cu)-Nb junction is shown on

the right-hand axis along with the temperature dependence of ΔH on the left-hand axis, shows a direct connection between quasiparticle current and ΔH. The ΔH values were extracted from V(H) curves measured at the respective temperatures with 200 µA current. **c** The voltage response of the Nb-(Pt(30 nm)/Cu)-Nb junction shows a ΔH of 4mT. Partial thinning of the Pt layer under the junction area in the same device did not change ΔH, as shown in panel (**d**). The false coloured FESEM image of the device and the schematic are inset in the respective panels, along with the IV curves.

This was verified in several junctions by milling out the Cu layer entirely from the junction region to realize Nb-Pt-Nb junctions from the same chips. A sample comparative response of junctions fabricated with and without the Cu layer in the barrier region on the same chip is shown in Fig. 3a. The temperature-dependent resistance of the junction with the Pt/Cu barrier showed a clear junction proximatization below 4 K, whereas the junction with only Pt barrier did not proximatize down to 2K. The inset current-voltage curves measured at 2K also show a clear supercurrent for the Nb-(Pt/Cu)-Nb junction, while Nb-Pt-Nb junction shows pure resistive behaviour. The energy dispersive X-ray (EDX) elemental maps of the actual devices are also shown in the inset of Fig. 3a. The absence of Cu in the junction region of the Nb-Pt-Nb junction compared to the Nb-(Pt/Cu)-Nb junction was clearly verifiable from these images. In order to directly rule out the contribution of the bulk Pt in the observed shift, we thinned down the Pt layer underneath a planar junction using focused ion beam milling to realize a suspended Nb-(Pt/Cu)-Nb junction with thinner Pt. The Fraunhofer patterns and the IV curves of the same device before and after thinning the Pt layer are compared in panels (c) and (d) in Fig. 3. The false colour FESEM images of the actual Josephson device and the schematic diagrams are shown as insets in the respective panels. In both cases, almost the same offset of ~4 mT was observed in the Fraunhofer patterns, as shown in the main panels of Fig. 3c, d. This observation confirms that there is a negligible bulk contribution in the observed effect. The inset IV curves in both cases, apart from a small change in the slope, did not show any change in the critical current. Consistent with the observation of Fig. 3a, no change in the critical current of the two

junctions reconfirms that Pt layer does not contribute to the supercurrent in the junction. The magnitude of the Rashba−Edelstein effect is expected to be directly proportional to the bias current. Therefore, the observation of seemingly current independent offset in the Fraunhofer pattern may appear contradictory to the above discussion. However, we note here that in these planar Josephson devices, the interface quasi-particle current $J_Q$ is responsible for the observed effect rather than the full bias current. In a Nb-(Pt/Cu)-Nb junction, measured at a fixed temperature of 2K, most of the bias current gets carried across the Josephson barrier through the Cu layer as a supercurrent for any bias current around the critical current. Therefore the magnitude of $J_Q$, responsible for the Rashba−Edelstein effect, does not get affected with the bias current, as seen in Fig. 2f. Since the spin moment $M_{RE}$ is directly proportional to the spin density $<\vec{\sigma}_{RE}>$ created by the Rashba−Edelstein effect[19] the magnitude of $J_Q$ can be estimated using a known value of the Rashba coefficient. A detailed representative calculation of $J_Q$ for a junction with ΔH ~ 10 mT has been shown in the Supplementary Note 7. Using a Rashba coefficient of 0.001 eV.Å for Pt/Cu interface[30] we obtained a very low quasiparticle current of the order of 1 nA at the Rashba interface. Considering the fact that the Rashba interface is a conduction path parallel to the Cu layer proximatized by the superconducting Nb electrodes, which carries most of the bias current across the junction, a low value of $J_Q$ is expected. We must mention here that an estimation of the Rashba coefficient $\alpha_R$, in this device geometry is feasible only with a microscopic model calculation which allows for an independent estimation of the quasiparticle current at the Pt/Cu interface.

## Discussion

Since the above discussion excludes the bulk contribution to the induced non-vanishing spin moment, the only possible source of a net moment in the junction barrier is, therefore, the spin-moment created by the Rashba–Edelstein effect at the Pt/Cu interface[28,50,56]. Cooper pair breaking at the interface of Pt and Cu[57,58] can result in a quasi-particle current at the interface, generating the Rashba–Edelstein effect and the consequent spin density[59–63]. The net shift in the Fraunhofer patterns ($\Delta H$) as a function of temperature, measured with a fixed bias current of 200 $\mu A$, also supports this argument. The temperature dependence of $\Delta H$ has been plotted on the left-hand axis of Fig. 3b. The junction resistance has been plotted on the right-hand axis on the same plot. This plot clearly shows that an increase in $\Delta H$ follows the pattern of a decrease in the junction resistance. In the R(T) data shown in Fig. 3b, the proximatization of the junction area (essentially the Cu barrier) starts below ~4.5 K. As the temperature decreases further, the magnitude of the supercurrent through the Cu barrier increases, leading to an increase in the quasiparticle current density $J_Q$ at the Pt/Cu interface. Since the magnitude of the Rashba–Edelstein effect is directly proportional to the bias current[13], the increase in $J_Q$ leads to the increase in the observed offset $\Delta H$ in the Fraunhofer pattern.

A closer look at the Fraunhofer patterns of planar junctions with Pt underlayer enunciates a peculiar asymmetry between the two parts of the pattern around the central minimum. A flux quantum through the junction equates to a relatively lower external field while decreasing the field magnitude compared to the case of increasing field magnitude. This general aspect of all the Nb-(Pt/Cu)-Nb junctions and SQUIDs has been emphasized in Fig. 4a where the V(H) curve has been plotted for a sample with 30 nm of Pt underlayer. In this figure, the arrows indicate the direction of the field sweep, and the solid lines are fitted to a Fraunhofer-type relation. The two halves of the

Fraunhofer pattern around the central minimum were fitted separately due to the clear asymmetry around the central minimum observed in the data. The fittings show that the external field corresponding to two flux quanta on both sides of the central minimum differ by 2 mT in this particular junction. A schematic sketch of the magnetic flux and current distribution through the junction at the maxima of the V(H) curve (corresponding to a net zero supercurrent in the junction) are also shown in Fig. 4a. In the presence of the out-of-plane external magnetic field, a component of the induced Rashba–Edelstein spin moment $M_{RE}$, along the direction of the magnetic field also adds a flux into the planar junction, as $M_{RE}$ is generated directly under the junction area[37]. It is important to note here that the quasi-particle current density at the Pt/Cu interface is directly proportional to the supercurrent density in the proximatized Cu barrier. Therefore, the screening current distribution in the junction due to the external magnetic field is coupled to the $M_{RE}$. When the magnitude of the external magnetic field is decreased, the junction opposes the changing magnetic field by creating screening currents in the negative direction, leading to the generation of a negative $M_{RE}$ in some parts of the junction. Consequently, the central minimum of the V(H) curve, corresponding to a net zero flux through the junction, appears at a positive field in the decreasing sweep of the magnetic field. Note that unwanted trapped flux in the direction of the applied magnetic field would shift the central minimum to the negative field region during the down-sweep. In order to verify this locking effect of $M_{RE}$ with the screening currents in the junction, we recorded the net offset $\Delta H$ by systematically varying the range of field sweep (defined as $H_{minor}$). A sample "minor loop" has been compared with the full sweep of the magnetic field in Fig. 4b. In this minor loop, the magnetic field was swept between $H_{minor} = \pm 6$ mT. The overall offset $\Delta H$ between the up-sweep and the down-sweep produced at the minimum of the V(H) curve has been plotted as a function of the

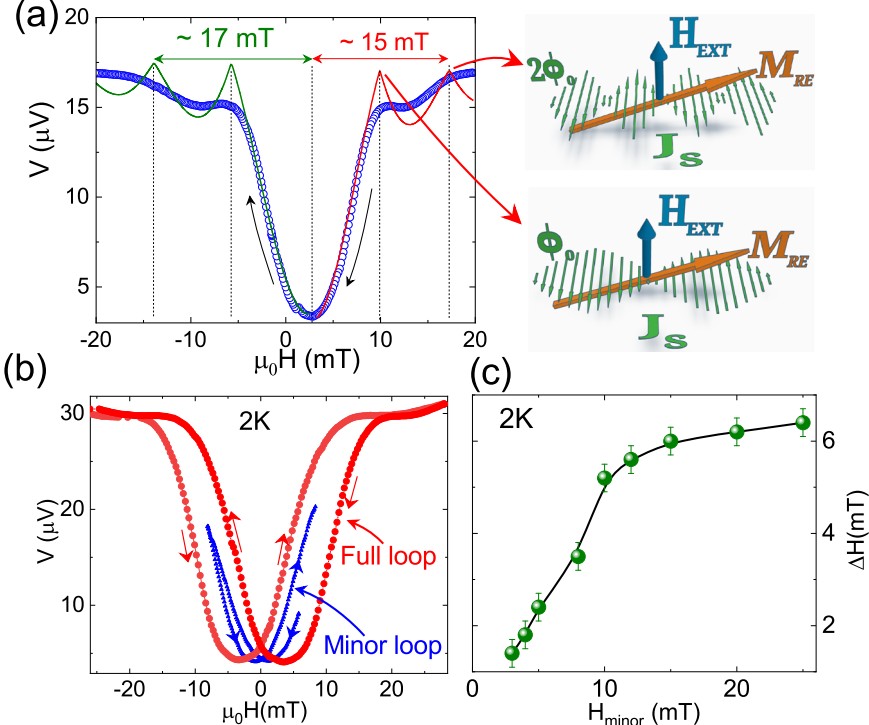

**Fig. 4 | Phase biasing effect in Nb-(Pt/Cu)-Nb JJ. a** V(H) curve of Nb-(Pt(30 nm)/Cu)-Nb junction fitted with Fraunhofer type relation shows a significant asymmetry about the central minimum. The open symbols are for experimental data points, measured at 2K with 100 $\mu A$ current, and the solid lines are the fits. Two sides about the central minimum were fitted separately. The current distribution in the junction with one flux quantum($\Phi_0$) and two flux quanta($2\Phi_0$) are shown in the inset

schematic, along with the non-equilibrium Rashba–Edelstein spin moment ($M_RE$). **b** Representative V(H) curves for an Nb-(Pt(30 nm)/Cu)-Nb junction depicting the major and minor field sweeps shows the phase biasing effect at zero field. The $\Delta H$ values obtained by varying the range of field sweeps in minor loops are plotted in panel (**c**) for the same device.

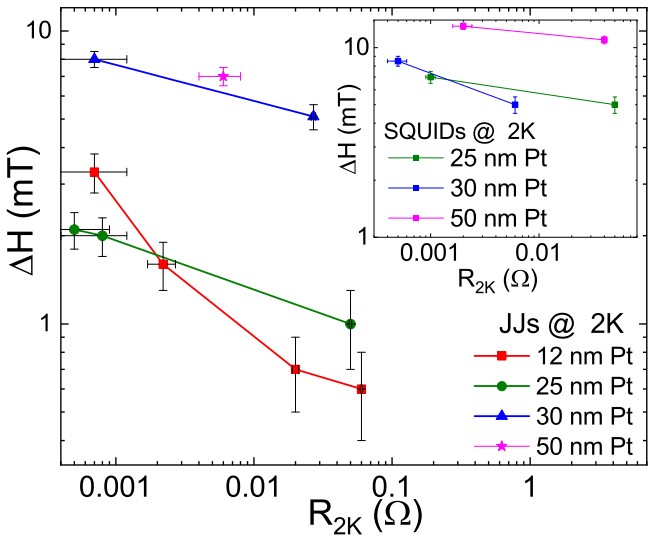

**Fig. 5 | Tuning of ΔH by device resistance.** Mapping of ΔH value with junction resistance at 2K for various JJs with Pt thicknesses of 12, 25, 30 and 50 nm. These values were extracted from V(H) measurements of the junctions biased at the respective critical currents. The solid lines are only guides to the eye. The inset shows the ΔH values as a function $R_{2K}$ for SQUID devices fabricated on the same chips. Some representative V(H) data are shown in the Supplementary Fig. 4.

sweeping range of minor field loops $H_{minor}$ in Fig. 4c. It shows that at the lower range of field sweep, there is a sharp rise in ΔH which is followed by a saturating tendency at higher range of field sweeps. In fact, the nature of the curve closely follows the V(H) curve. We also note that the reversal of spin moment $M_{RE}$ does not affect the observed value of ΔH because, in both cases, the spin moment $M_{RE}$ locks to the flux quantum only in the direction of the applied field, as shown in the schematic illustration in the Supplementary Fig. 7. In the Supplementary Note 6 we have demonstrated that reversal of spin moment does not alter the value ΔH, though the magnitude of $\alpha_R$ does change the magnitude of ΔH, as expected.

This observation indicates that controlling the amount of the superconducting screening current in the junction could lead to tunable offset ΔH, which amounts to tuning the effective phase of the junction at zero field. One of the major device parameters which can tune the magnitude of the screening currents in the planar Josephson junctions is the junction resistance. The geometrical dimensions of the junction and the degree of proximatization of the barrier are the primary factors which define the junction resistance. In Fig. 5, we show a collection of the ΔH values measured at 2K for several JJs and SQUIDs as a function of $R_{2K}$. For junctions with the same thickness of Pt layer, the $R_{2K}$ was varied by changing the geometrical dimensions of the junction. For example, the Nb-(Pt/Cu)-Nb junctions with 10 nm of Pt, reported in Fig. 5, were fabricated on the same chip by systematically increasing the milling depth of the Cu layer, which caused an increase in the junction resistance at 2K. Consequently, a systematic drop in the ΔH was observed with increasing junction resistance, as shown in Fig. 5. Junctions with other thicknesses of Pt under layer also showed a similar decreasing trend in the ΔH value with increasing $R_{2K}$. The ΔH value for SQUID devices as a function of $R_{2K}$, are shown in the inset of Fig. 5. Although the exact dependence of the ΔH on the junction resistance requires microscopic modelling, the experimental data clearly shows a consistent decrease in ΔH with increasing junction resistance in JJs and SQUID devices.

In summary, we have demonstrated the planar Josephson effect as a simple tool for direct detection of the equivalent magnetic strength of the non-equilibrium spin density created by the Rashba–Edelstein effect. We show that the spin moment can be easily coupled to the

Josephson junction by an external magnetic field, which leads to a distinct shift (ΔH) in the Fraunhofer pattern of the junction. From the dependence of ΔH on the junction resistance, it was found that quasi-particle current at the Pt/Cu interface in the junction barrier region was the primary tuning parameter of the observed effect. The breaking of both spatial inversion symmetry and time reversal symmetries provides the necessary conditions for realizing a phi-phase Josephson junction[64,65]. Our experiment shows that non-equilibrium spin moment coupled to planner Josephson devices can be a convenient way of attaining arbitrary phase in Josephson junctions via application of out of plane magnetic fields. A junction producing an on-demand initial phase using the Rashba–Edelstein effect could be very useful for magnetic-field-controlled phase biasing of superconducting quantum circuits.[66,67]

## Methods

### Multilayer deposition
The series of trilayer Pt/Cu/Nb films and bilayer Cu/Nb films were deposited on cleaned $Si/SiO_2$ substrates using DC magnetron sputtering of high purity Nb, Pt, and Cu metal targets. A 5 nm ultra-thin adhesion layer of Nb was used for the bilayer case as Cu has a very poor adhesion to the $SiO_2$. In all cases the Nb and Cu layers were kept fixed at 150 nm and 100 nm, respectively. The Pt layer thickness was varied across the series from 10 nm to 80 nm. Pt layer thickness was calibrated using X-ray reflectivity measurements as shown in the Supplementary Fig. 1. Prior to device fabrication 2 µm wide tracks of bilayer and trilayer samples were obtained by depositing films on lithographically patterned substrates and lifting-off.

### Junction fabrication
The lithographically patterned tracks were subsequently narrowed down by focused beam of Gallium ion using a Zeiss Crossbeam 340 system. We used 100 pA ion current at 30 KV for the initial narrowing of the track down to 500 nm and 10 pA current at 30 KV for narrowing the track width down to 300 nm and sidewall polishing to have a clean junction interface. The planar junction separation of the top Nb layer was realized by milling down from the top with a current of 5pA at 30 KV. For the device shown in the Fig. 3d the Pt layer under the planar JJ was thinned by milling at an angle of 89 degrees with the sample normal. The junction area were examined using EDAX mapping to ensure no leftover Nb in the junction area. Multiple planar Josephson junctions and SQUIDs were fabricated following the same protocol and examined using EDAX for Nb leftovers. It is important to mention here that the junction resistance in the planar junction geometry, varies directly with the separation between the Nb electrodes (L) and inversely with the width of track (d)[see supplementary Fig. 3]. The only control parameter for the top milling of the planar junction area is the exposure time of the ion beam. Therefore, in the process of ensuring no Nb leftover in the junction area, some Cu also gets etched out leading to a small variation in the Cu thickness from junction to junction. It is, therefore, inevitable to have variations of resistance at 2K from junction to junction fabricated even on the same chip.

### Transport measurements
The resistance of the planar JJ and SQUID devices were measured in four probe arrangements with a magnetic field perpendicular to the plane of the devices. A liquid cryogen free low-temperature cryostat, equipped with a precision low magnetic field measurement option, was used for all the measurements. The general features of the Josephson devices are discussed in the supplementary Fig. 2 in detail. The magnetic field dependence of the junction resistance (Fraunhofer patterns) were measured with magnetic field perpendicular to the planar devices, unless specified otherwise. The remnant magnetic field of the system was subtracted from all V(H) data.

## Data availability

The data used in this paper are available from the authors upon request.

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

## Acknowledgements

The authors would like to thank the National Institute of Science Education and Research (NISER), Department of Atomic Energy, Government of India, for funding the research work through project number RIN-4001. We would like to acknowledge help from Ritarth Chaki for making the graphics in Fig. 1a, b.

## Author contributions

T.S. prepared the multilayer films and fabricated the devices. T.S. carried out the experiments along with A.K. T.S. and K.S. analyzed the data and prepared the manuscript.

## Competing interests

The authors declare no competing interests.
