## [Peer Review File · Nature Communications]

Phase biasing of a Josephson junction using Rashba-Edelstein effectREVIEWER COMMENTS

Reviewer #1 (Remarks to the Author):

The authors have observed hysteretic Fraunhofer interference patterns in Nb-(Pt/Cu)-Nb planar Josephson junctions (JJs) when an out-of-plane (OOP) magnetic field is applied to the (Cu/Pt) Rashba-type interface. The observed hysteretic interference patterns are attributed to effective locking of the applied -OOP-field-induced screening supercurrents with the non-equilibrium spin population generated by the Rashba-Edelstein (RE) effect. They have also shown that via modulating the effective resistance of the Nb-(Pt/Cu)-Nb JJs, a distinctive shift (ΔH) in the Fraunhofer interference pattern, i.e. the experimental signature of RE-effect-driven phase bias through screening supercurrents to the JJs, can be tuned.

While I find the results quite interesting, there are a few issues that must be addressed before further consideration of this manuscript in *Nature Communications*.

1) Basically, the authors have measured the averaged voltage V as a function of OOP magnetic field $\mu_0 H_{OOP}$ for Nb-(Pt/Cu)-Nb JJs (Fig. 2c) and SQUIDs (Fig. 2d) rather than performing the current-voltage I-V measurements at different applied OOP fields in detail.

It would be great if the authors could compare the entire OOP magnetic-field interference patterns $I_c(\mu_0 H_{OOP})$ for both sweep-up and sweep-down $\mu_0 H_{OOP}$ with the $V(\mu_0 H_{OOP})$ data to see directly whether the hysteretic behaviours are connected to each other in an inverse manner. The associated measurement and analysis can be found in recent publications [Nat. Commun. 5, 4771 (2014), Phys. Rev. Lett. 116, 077001 (2016), Nat. Nanotechnol. (2023). <https://doi.org/10.1038/s41565-023-01336-z>].

2) In the section of Generation of spin density at the Cu/Pt interface (on page 6), the authors state that 'In the present case, while sweeping the magnetic field from negative field, the central peak of the Fraunhofer pattern appears in the negative field regime. Since there are no magnetic moments in the present case, the observed shift can arise only from a net spin-moment present in the junction. Fig 2 (d) shows that there is no relative shift in the Fraunhofer patterns in a junction without the Pt layer. Therefore, the presence noncentrosymmetric potential gradient in the Pt layer must be the origin of spin moment in the planar junction [47, 48]. ...'

In this regard, I wonder if the Cu/Pt bilayer replaced with either a Pt/Cu inverted or a Cu/Ta bilayer (equivalently, the Rashba spin-splitting near the interface is reversed) [PRB 102, 144415 (2020)], which difference will happen to the hysteretic patterns (Fig. 2c,d and Fig. 3c,d)? I think that from this additional experiment, the authors can further support that the non-equilibrium spin population created by the interface RE (or inverse spin-galvanic) effect is responsible for the distinctive shift (ΔH).

3) Based on the junction-resistance-dependent ΔH data (Fig. 5), the authors should try to estimate the Rashba coefficient α_R , quantifying how much large the non-equilibrium spin population (JQ) is created at the Cu/Pt interface for a given bias current, and to compare it with the previous study of FMR spin-pumping and ISHE [PRB 102, 144415 (2020)]. I think this discussion needs to be added in the revised manuscript.

4) As in prior studies [Nat. Phys. 12, 568–572 (2016), Nat. Commun. 10, 126 (2019), Nat. Commun. 11, 212 (2020), Nat. Nanotechnol. (2023). <https://doi.org/10.1038/s41565-023-01336-z>], I would suggest the authors investigate whether the phase-biasing is indeed present in the Nb-(Pt/Cu)-Nb JJs by closely looking at the SQUID data (Fig. 2d, Fig. A4 and Fig. A5) with sweeping $\mu_0 H_{OOP}$ up and down, in particular around the zero-field.

I hope it helps improve the overall quality of the paper to be more suited for the journal of *Nature Communications*.

Response to Referee comments

Referee comment (1) The authors have observed hysteretic Fraunhofer interference patterns in Nb-(Pt/Cu)-Nb planar Josephson junctions (JJs) when an out-of-plane (OOP) magnetic field is applied to the (Cu/Pt) Rashba-type interface. The observed hysteretic interference patterns are attributed to effective locking of the applied -OOP-field-induced screening supercurrents with the non-equilibrium spin population generated by the Rashba-Edelstein (RE) effect. They have also shown that via modulating the effective resistance of the Nb-(Pt/Cu)-Nb JJs, a distinctive shift (ΔH) in the Fraunhofer interference pattern, i.e. the experimental signature of RE-effect-driven phase bias through screening supercurrents to the JJs, can be tuned.

While I find the results quite interesting, there are a few issues that must be addressed before further consideration of this manuscript in Nature Communications.

Author Response: We thank the referee for a careful review of our manuscript and for the constructive suggestions to improve it. We address all the points raised by the referee in the discussions below.

Referee comment (2) Basically, the authors have measured the averaged voltage V as a function of OOP magnetic field $\mu_0 H_{OOP}$ for Nb-(Pt/Cu)-Nb JJs (Fig. 2c) and SQUIDs (Fig. 2d) rather than performing the current-voltage I - V measurements at different applied OOP fields in detail. It would be great if the authors could compare the entire OOP magnetic-field interference patterns $I_c(\mu_0 H_{OOP})$ for both sweep-up and sweep-down $\mu_0 H_{OOP}$ with the $V(\mu_0 H_{OOP})$ data to see directly whether the hysteretic behaviours are connected to each other in an inverse manner. The associated measurement and analysis can be found in recent publications [Nat. Commun. 5, 4771 (2014), Phys. Rev. Lett. 116, 077001 (2016), Nat. Nanotechnol. (2023), <https://doi.org/10.1038/s41565-023-01336-z>].

Author Response: We fully acknowledge the significance of comparing the sweep up and sweep down voltage data with the Fraunhofer pattern obtained from the $I_c(H)$ (critical current-magnetic field) measurements. Accordingly, we have now measured the $I_c(H)$ of a junction (I_c extracted from I - V measurements at various magnetic fields) during the up and down sweeps of the magnetic field as plotted in the left hand side axis of the following figure. The corresponding $V(H)$, measured at a fixed bias current close to the critical current, is plotted in the right hand side axis. We see that the magnitude of the offset between up-sweep and down-sweep data in both measurements match very well with each other. This observation verifies the equivalence of both modes of measurements. We have now included the following comparative figure into the supplementary information section in the revised manuscript as Supplementary note 5.

Referee comment (3) In the section of Generation of spin density at the Cu/Pt interface (on page 6), the authors state that ‘In the present case, while sweeping the magnetic field from negative field, the central peak of the Fraunhofer pattern appears in the negative field regime. Since there are no magnetic moments in the present case, the observed shift can arise only from a net spin-moment present in the junction. Fig 2 (d) shows that there is no relative shift in the Fraunhofer patterns in a junction without the Pt layer. Therefore, the presence noncentrosymmetric potential gradient in the Pt layer must be the origin of spin moment in the planar junction [47, 48]....’

In this regard, I wonder if the Cu/Pt bilayer replaced with either a Pt/Cu inverted or a Cu/Ta bilayer (equivalently, the Rashba spin-splitting near the interface is reversed) [PRB 102, 144415 (2020)], which difference will happen to the hysteretic patterns (Fig. 2c,d and Fig. 3c,d)? I think that from this additional experiment, the authors can further support that the non-equilibrium spin population created by the interface RE (or inverse spin-galvanic) effect is responsible for the distinctive shift (ΔH).

Author Response: The work cited by the referee [PRB 102, 144415 (2020)] demonstrated that in the ultrathin limits, the sign of the spin polarization can be changed by reversing the stacking order at a Rashba interface. Changing stacking order reverses the direction of the interfacial electric field at the Rashba interface, which eventually leads to the reversal of spin-polarization direction. The reversal of spin-polarization in this experiment was measured via the spin-pumping voltage across the sample which showed a clear sign reversal. The question is whether a polarization reversal in our experimental geometry would lead to any change in the observed ΔH ? This is a very pertinent question by the referee which, we believe, should have been discussed.

We have now added the following detailed discussion to a separate supplementary section as supplementary note 6, to address this question.

→The basic architecture of our device is to position a planar Josephson junction just above a region with an in-plane non-equilibrium spin density. The applied out-of-plane (OOP) magnetic field pulls the spin-moment out of the Rashba interface which contributes to the total flux through the junction. In this process the spin-moment gets locked to the flux quantum and the rigidity of the flux quantum leads to the squeezing of the Fraunhofer pattern (Fig 4(a) in the main text) during the down sweep of the OOP magnetic field. As a result, the junction retains a finite phase even when the external OOP magnetic field is set back to zero. In the following schematic we compare two scenarios where the directions of the Rashba-Edelstein spin moment M_{RE} have been reversed (either by changing stacking order or by using materials with opposite spin Hall angles in the two cases). In both cases, the spin moment locks to the flux quantum only in the direction of the applied field. Therefore, in both cases, we can expect a squeezing of the Fraunhofer pattern during the down sweep of the field only. Hence the presented technique is not sensitive to the direction of the induced non-equilibrium spin

moment, but only to the magnitude of the spin-moment. In order to verify this argument we have conducted the following two control experiments.

(i) Effect of reversing direction of spin-polarization on ΔH :

Under the Rashba-Edelstein model the non-equilibrium spin expectation value can be written as

$$\frac{\langle \vec{\sigma} \rangle}{A} = \frac{\alpha_R m \hbar}{|e|(\alpha_R^2 m + \hbar^2 \epsilon_F)} \left[\vec{e}_z \times \vec{J}_c \right] \quad (1)$$

where (\vec{e}_z) is the unit vector along the interface electric field and (\vec{J}_c) is the surface current density at the Rashba interface [Johansson et al., Phys Rev B 93, 195440 (2016)]. Clearly, the direction of polarization of the induced spin-moment can be reversed either by reversing the interface electric field (changing stacking order as suggested by the referee) or by changing the direction of the interface current. In our experimental geometry, changing the stacking order of (Pt/Cu) to (Cu/Pt) is not feasible due to the fact that the proximatization length of the Ga ion poisoned (due to FIB milling) Pt layer is extremely small. Therefore, a planar Josephson junction with a barrier gap of 50-100 nm (gap limited by the technique of FIB milling) can not be realized by having Pt as the top layer in the barrier region. From the above expression, the alternative route to reverse the direction of spin-polarization is to change the direction of bias current in the junction. From the above expression, an alternative route to reverse the direction of spin-polarization is to change the direction of bias current in the junction. In the following figure we compare the $V(H)$ responses of a junction, measured with positive and negative bias currents, keeping the direction of magnetic field unchanged. As evident from this figure, ΔH does not suffer any change by the reversal of spin-polarization, consistent with the argument presented in the previous paragraph.

(ii) Effect of spin-orbit coupling strength:

The magnitude of the spin moment generated due to the Rashba-Edelstein effect is proportional to the spin-orbit coupling strength of the heavy metal. Therefore, replacing Pt with another material is expected to change the value of ΔH in our experiment. We have performed this control experiment by replacing Pt with Nb, which has a much lower spin-orbit coupling strength. The thickness of Nb was ~ 10 nm, which is much less than the coherence length of Nb for the film to become superconducting at 2K. We show the $V(H)$ curve of such a junction measured at 2 K in the following figure. Compared to a junction with Pt underlayer, we find a significantly lower ΔH for the case of Nb underlayer.

Referee comment (4) Based on the junction-resistance-dependent ΔH data (Fig. 5), the authors should try to estimate the Rashba coefficient α_R , quantifying how much large the non-equilibrium spin population (JQ) is created at the Cu/Pt interface for a given bias current, and to compare it with the previous study of FMR spin-pumping and ISHE [PRB 102, 144415 (2020)]. I think this discussion needs to be added in the revised manuscript.

Author Response: Following the suggestion of the referee we have tried to estimate the value of J_Q (the interfacial quasiparticle current) causing the shift ΔH in our junctions using Rashba constant α_R from the literature. The following detailed discussion has now been included as a separate supplementary section, supplementary note 7. A shorter version of the discussion has also been added to the revised manuscript.

→ A phase shift in the Fraunhofer pattern of a Josephson junction can occur only in presence of an additional flux. In our experiment the observed shift in the Fraunhofer patterns (ΔH) is caused by the flux due to the spin moment M_{RE} created by the Rashba-Edelstein effect at the interface, which can be written as

$$M_{RE} = \mu_0(\Delta H/2)$$

The factor of 2 comes from the fact that ΔH in the manuscript has been defined as the total shift between the up-sweep and down-sweep Fraunhofer patterns. The origin of the spin moment M_{RE} is the average non-equilibrium spin $\langle \sigma_{RE} \rangle$. Therefore, M_{RE} can also be written as

$$M_{RE} = g\left(\frac{e}{2m}\right) \langle \sigma_{RE} \rangle$$

Where g , e , and m are the gyromagnetic ratio, charge and mass of the electron, respectively. Therefore we can write

$$\langle \sigma_{RE} \rangle = \frac{\mu_0 \Delta H}{2} \left(\frac{m}{e}\right)$$

Using this expression we can directly relate the experimentally observed parameter ΔH to the magnitude of interfacial quasiparticle current density as

$$\frac{1}{A} \left(\frac{\mu_0 \Delta H}{2}\right) = \frac{\alpha_R \hbar}{(\alpha_R^2 m + \hbar^2 \varepsilon_F)} J_Q$$

Here A is the junction area, ε_F is the Fermi energy of Pt at the Pt/Cu interface. For the purpose of an estimation we consider the following $V(H)$ data for a planar Nb-(Pt/Cu)-Nb junction which was measured with a bias current of $100 \mu A$. Using the following values

$$\frac{\mu_0 \Delta H}{2} = 50 \text{ Oe} \approx 3979 \text{ A/m}$$

$$A = 50 \text{ nm} \times 300 \text{ nm} = 1.5 \times 10^{-14} \text{ m}^2$$

$$\varepsilon_F \approx 1.4 \times 10^{-18} \text{ J}$$

$$\alpha_R = 0.001 \text{ eV} \cdot \text{\AA} \approx 1.6 \times 10^{-32} \text{ J} \cdot \text{m}$$

a quasiparticle surface current of 0.7 nA was estimated at the Pt/Cu interface for this particular junction. Considering the fact that the Cu layer, proximatized by the superconducting Nb electrodes, carries all the bias current, the low value of the interface quasiparticle current is not unexpected. On the other hand, the direct estimation of α_R is not feasible in our case without microscopic calculations of pair breaking effects at the Pt/Cu interface. This is because it is not feasible to estimate the value of J_0 from the measured parameters of the experiment independently.

Referee comment (5) As in prior studies [Nat. Phys. 12, 568–572 (2016), Nat. Commun. 10, 126 (2019), Nat. Commun. 11, 212 (2020), Nat. Nanotechnol. (2023). <https://doi.org/10.1038/s41565-023-01336-z>], I would suggest the authors investigate whether the phase-biasing is indeed present in the Nb-(Pt/Cu)-Nb JJs by closely looking at the SQUID data (Fig. 2d, Fig. A4 and Fig. A5) with sweeping $\mu_0 H_{OOP}$ up and down, in particular around the zero-field.

Author Response: In the references cited by the referee, the authors have primarily employed the asymmetric SQUID technique to measure current-phase relationship (CPR) for a junction under test. Such measurements require the reference junction of a SQUID to have much higher critical current compared to the junction under test, so that the phase of the reference junction remains practically unchanged in small fields. This allowed the authors to track the phase evolution of the test junction reliably, with respect to the reference junction.

We emphasize the fact that the SQUID devices in our experiment are symmetric in geometry and they do not have a reference junction. Therefore, a similar analysis of the low field SQUID data, in our case, is not meaningful. In addition, the large phase biasing of the individual junctions dominates over any phase biasing of the SQUID. Following the suggestions of the referee we have now carefully analyzed the low field regions of the SQUID data in all devices. In the following figure, we have shown magnified views of the $V(H)$ data for two different SQUID devices (with different thickness of Pt layers) near zero field. Although we do notice some deformation in the sinusoidal SQUID oscillations, they are not identifiable consistently in all the oscillations, both in the up-sweep and in the down-sweep data.

Although it is possible to fabricate asymmetric SQUID devices using the same FIB milling technique used in this manuscript, the large phase bias from the individual junctions is expected to hinder a

reliable CPR measurement. This is because of the fact that the phase of the larger reference junction of the asymmetric SQUIDs, fabricated by the FIB technique, does not remain fixed during the field sweep due to the Rashba-Edelstein effect underneath. In order to experimentally verify this statement we

fabricated a few asymmetric SQUIDs from Pt(50 nm)-Cu-Nb trilayer following the same process that we used for all other devices, and studied the $V(H)$ behavior. The ratio of the junction widths for these asymmetric SQUIDs was 1:5 (200 nm: 1 μm), which ensured a much higher critical current of the larger junction, following the references cited by the referee. However, the magnified view of the low field section of the $V(H)$ data, plotted in the following figure, did not show systematic deformations in the sinusoidal SQUID oscillations near zero field, as expected in a true current-phase relation measurement. Therefore, conclusive information on the current-phase relation

of the smaller junction was not possible to obtain in this device geometry.

We thank the referees for very constructive suggestions which have certainly improved the clarity of our manuscript.

REVIEWERS' COMMENTS

Reviewer #1 (Remarks to the Author):

The authors have clearly improved their manuscript in response to my comments. I would thus recommend the revised manuscript for publication in *Nature Communications* with few final suggestions below.

Even though there is no magnetic material in the Nb-(Pt/Cu)-Nb planar Josephson junctions studied, the claimed RE-effect-driven phase bias seemingly appears only when screening supercurrents under application of an OOP external magnetic field are induced and effectively locked with the non-equilibrium spin population generated by the RE effect.

To avoid any possible confusion for non-specialist readers, some statements in 1) Abstract and 2) Conclusion section need to be re-written as follows.

1) Abstract:

'This platform also offers a magnetic-field-controlled phase biasing mechanism in conjunction with the RE spin-orbit effect for superconducting quantum circuits.'

2) Conclusion:

Please remove or rewrite the statement of 'In our device geometry, \sim the time-reversal symmetry.' Here, I do not think that the time-reversal symmetry necessary for lateral Josephson supercurrent non-reciprocity is broken. Please refer to a recent paper [Nature Materials 21, 1008 (2022)] for technical details.

'Our experiment shows that non-equilibrium spin moment coupled to planar Josephson devices can be a convenient way of attaining arbitrary phase in Josephson junctions via application of OOP magnetic fields.'

'A junction producing an on-demand initial phase using the RE effect could be very useful for magnetic-field-controlled phase biasing of superconducting quantum circuits.'

Reviewer #2 (Remarks to the Author):

I have studied the manuscript entitled 'Phase biasing of a Josephson junction using Rashba-Edelstein effect, by Senapati et al., submitted to *Nature Communications*. I find the research question very interesting, and the data and the analysis clear. I can follow the way the authors come to their conclusions, and I agree with them. I am happy to advise publication.

The idea behind the experiments is quite straightforward, and uses known physics, but in a novel way. That idea is to probe the well-known (Edelstein-)Rashba effect, that should lead to a certain amount of spin polarization S at the Rashba interface, by building it into a Josephson junction. The magnetization from S shifts the maximum in the critical current (I_c) of the Fraunhofer pattern away from zero. The samples used are of the Nb / (Cu/Pt) / Nb type, in which the spin-orbit coupler Pt generates S . What I find very convincing is that the I_c peak occurs before (after) zero field when sweeping the applied field up (down), which is opposite to what is seen in ferromagnets, as the authors point out correctly. An interesting subtlety is that enough quasiparticle current has to flow to the Pt to generate S . The authors argue convincingly that that is the case.

I also studied the rebuttal of the authors to the comments of a referee of an earlier version. Interestingly, the referee touched upon exactly those points I would also have had questions

about. The Supplement now gives both $V(H)$ and $I_c(H)$. The authors explain why they cannot revert the sequence of Cu and Pt, which I find reasonably convincing, and they give the results of inverting the bias current, which is helpful. They make a reasonable estimate for α_R , and they endeavoured to answer the question on SQUID data as well as they could. In my opinion, they responded very well, and I cannot see loose end.

In conclusion, I would advise to publish this interesting work in *Nature Communications*.

Response to Referee comments

Reviewer #1:(Remarks to the Author)

Referee comments: The authors have clearly improved their manuscript in response to my comments. I would thus recommend the revised manuscript for publication in Nature Communications with few final suggestions below.

Author Response: We thank the referee for a careful review and the recommendation for publication of our manuscript.

Referee comments: Even though there is no magnetic material in the Nb-(Pt/Cu)-Nb planar Josephson junctions studied, the claimed RE-effect-driven phase bias seemingly appears only when screening supercurrents under application of an OOP external magnetic field are induced and effectively locked with the non-equilibrium spin population generated by the RE effect.

To avoid any possible confusion for non-specialist readers, some statements in 1) Abstract and 2) Conclusion section need to be re-written as follows.

1) Abstract:

'This platform also offers a magnetic-field-controlled phase biasing mechanism in conjunction with the RE spin-orbit effect for superconducting quantum circuits.'

Author Response: We agree with the referee and in the revised manuscript we have modified the sentence in abstract as per suggestion.

Referee comments:

2) Conclusion:

Please remove or rewrite the statement of 'In our device geometry, \sim the time-reversal symmetry.' Here, I do not think that the time-reversal symmetry necessary for lateral Josephson supercurrent non-reciprocity is broken. Please refer to a recent paper [Nature Materials 21, 1008 (2022)] for technical details.

Author Response: We have removed the refereed sentence from the manuscript as per the suggestion of the referee, for clarity.

Referee comments: 'Our experiment shows that non-equilibrium spin moment coupled to planar Josephson devices can be a convenient way of attaining arbitrary phase in Josephson junctions via application of OOP magnetic fields.'

'A junction producing an on-demand initial phase using the RE effect could be very useful for magnetic-field-controlled phase biasing of superconducting quantum circuits.'

Author Response: The sentences have been revised as per the suggestion of the referee.

Reviewer #2 (Remarks to the Author):

Referee comments: I have studied the manuscript entitled 'Phase biasing of a Josephson junction using Rashba-Edelstein effect, by Senapati et al., submitted to Nature Communication. I find the research question very interesting, and the data and the analysis clear. I can follow the way the authors come to their conclusions, and I agree with them. I am happy to advise publication.

The idea behind the experiments is quite straightforward, and uses known physics, but in a novel way. That idea is to probe the well-known (Edelstein-)Rashba effect, that should lead to a certain amount of spin polarization S at the Rashba interface, by building it into a Josephson junction. The magnetization from S shifts the maximum in the critical current (I_c) of the Fraunhofer pattern away from zero. The samples used are of the Nb / (Cu/Pt) / Nb type, in which the spin-orbit coupler Pt generates S . What I find very convincing is that the I_c peak occurs before (after) zero field when sweeping the applied field up (down), which is opposite to what is seen in ferromagnets, as the authors point out correctly. An interesting subtlety is that enough quasiparticle current has to flow to the Pt to generate S . The authors argue convincingly that that is the case.

I also studied the rebuttal of the authors to the comments of a referee of an earlier version. Interestingly, the referee touched upon exactly those points I would also have had questions about. The Supplement now gives both $V(H)$ and $I_c(H)$. The authors explain why they cannot revert the sequence of Cu and Pt, which I find reasonably convincing, and they give the results of inverting the bias current, which is helpful. They make a reasonable estimate for α_R , and they endeavored to answer the question on SQUID data as well as they could. In my opinion, they responded very well, and I cannot see loose end.

In conclusion, I would advise to publish this interesting work in Nature Communications.

Author Response: We thank the referee for a careful review, for the encouraging remarks and for the recommendation for publication of our manuscript.